# Unilateral Hypofunction of the Masseter Leads to Molecular and 3D Morphometric Signs of Atrophy in Ipsilateral Agonist Masticatory Muscles in Adult Mice

**DOI:** 10.3390/ijms241914740

**Published:** 2023-09-29

**Authors:** Julián Balanta-Melo, Andrea Eyquem-Reyes, Noelia Blanco, Walter Vásquez, Kornelius Kupczik, Viviana Toro-Ibacache, Sonja Buvinic

**Affiliations:** 1School of Dentistry, Faculty of Health, Universidad del Valle, Cali 760043, Colombia; julian.balanta@correounivalle.edu.co; 2Department of Anatomy, Cell Biology & Physiology, Indiana University School of Medicine, Indianapolis, IN 46202, USA; 3Indiana Center for Musculoskeletal Research, Indiana University School of Medicine, Indianapolis, IN 46202, USA; 4Institute for Research in Dental Sciences, Faculty of Dentistry, Universidad de Chile, Santiago 8380544, Chile; andrea.eyquem@ug.uchile.cl (A.E.-R.); noeliaa.91@gmail.com (N.B.); wavasqueza@gmail.com (W.V.); mtoroibacache@odontologia.uchile.cl (V.T.-I.); 5Department of Anthropology, Faculty of Social Sciences, Universidad de Chile, Santiago 7750000, Chile; 6Max Planck Institute for Evolutionary Anthropology, 04103 Leipzig, Germany; 7Center for Exercise, Metabolism and Cancer Studies CEMC2016, Faculty of Medicine, Universidad de Chile, Santiago 8380453, Chile

**Keywords:** masticatory muscles, X-ray microtomography, muscular atrophy, botulinum toxins, type A

## Abstract

Mice are commonly used to study mandibular dynamics due to their similarity in chewing cycle patterns with humans. Adult mice treated unilaterally with botulinum toxin type A (BoNTA) in the masseter exhibit atrophy of this muscle characterized by an increase in the gene expression of atrophy-related molecular markers, and a reduction in both muscle fiber diameter and muscle mass at 14d. However, the impact of this muscle imbalance on the non-treated masticatory muscles remains unexplored. Here, we hypothesize that the unilateral masseter hypofunction leads to molecular and 3D morphometric signs of atrophy of the masseter and its agonist masticatory muscles in adult mice. Twenty-three 8-week-old male BALB/c mice received a single injection of BoNTA in the right masseter, whereas the left masseter received the same volume of saline solution (control side). Animals were euthanized at 2d, 7d, and 14d, and the masticatory muscles were analyzed for mRNA expression. Five heads were harvested at 14d, fixed, stained with a contrast-enhanced agent, and scanned using X-ray microtomography. The three-dimensional morphometric parameters (the volume and thickness) from muscles in situ were obtained. *Atrogin-1/MAFbx*, *MuRF-1,* and *Myogenin* mRNA gene expression were significantly increased at 2 and 7d for both the masseter and temporalis from the BoNTA side. For medial pterygoid, increased mRNA gene expression was found at 7d for *Atrogin-1/MAFbx* and at 2d–7d for *Myogenin*. Both the volume and thickness of the masseter, temporalis, and medial pterygoid muscles from the BoNTA side were significantly reduced at 14d. In contrast, the lateral pterygoid from the BoNTA side showed a significant increase in volume at 14d. Therefore, the unilateral hypofunction of the masseter leads to molecular and morphological signs of atrophy in both the BoNTA-injected muscle and its agonistic non-injected masticatory muscles. The generalized effect on the mouse masticatory apparatus when one of its components is intervened suggests the need for more clinical studies to determine the safety of BoNTA usage in clinical dentistry.

## 1. Introduction

The specialized craniofacial skeletal muscles (i.e., masticatory muscles) control the mandibular movements to perform vital functions such as mastication in mammals [1,2]. Mice, as in humans, exhibit a main group of masticatory muscles, including the masseter, the temporalis, the medial pterygoid, and the lateral pterygoid [3]. Considering that mice share some similarities with humans in the chewing cycle, such as the opening/closing phases, these animals have been commonly used as preclinical models to study mandibular dynamics [1,2,4,5]. In adult mice, the masseter is the biggest masticatory muscle, and stabilizes the mandible during mastication while moving from back to front for food processing using the molar teeth. To determine the masseter input, in a previous study, it was bilaterally paralyzed using a targeted botulinum toxin type A (BoNTA) intramuscular injection [2].

BoNTA is a neurotoxin that specifically reaches the neuromuscular junction in the skeletal muscle, leading to a transient blockade in the release of acetylcholine and, therefore, transitory muscle hypofunction [6,7]. In previous research published by our group, the unilateral masseter hypofunction, induced by BoNTA, resulted in increased mRNA levels of atrophy-related molecular markers, atrogenes (*Atrogin-1/MAFbx*, *MuRF-1*), and *Myogenin* (a neurogenic atrophy-inductor) at 7d post-intervention, which was followed by a reduction in both muscle fiber diameter and muscle mass at 14d [8,9]. The early molecular and microstructural response to BoNTA intervention occurs only in the treated masseter, but not in its contralateral control which is injected with saline solution [8,9]. In addition, indirect evidence based on the mouse molar tooth wear suggests that the BoNTA-injected side exhibits reduced masticatory function in mice treated unilaterally [10]. However, the impact of the unilateral masseter hypofunction on the gene expression and phenotype of the non-treated masticatory muscles, either from the ipsilateral or contralateral side, remains unexplored.

The mechanism of action of BoNTA, and its impact on skeletal muscle function and shape, has gained attention in clinical dentistry to manage pathological and aesthetic craniofacial conditions [11]. Temporomandibular disorders (TMDs) are the most prevalent musculoskeletal diseases that affect the masticatory system, compromising vital functions such as mastication and speaking at physical and psychosocial levels [12,13,14], and reach as high as 31% in the adult population [15]. Furthermore, due to the COVID-19 pandemic, there was an increase in the prevalence of TMDs among patients affected by this respiratory disease [16,17]. Importantly, public health strategies to control COVID-19 spread, such as the use of surgical masks by the overall population, may impact the electromyographic activity of masticatory muscles such as the masseter and the temporalis, with differences between healthy patients and those affected by TMDs [18]. In particular, electromyography of masticatory muscles has been proposed as a potential diagnostic tool to identify pain-related TMDs such as myofascial pain [13]. Thus, considering the benefits of managing painful TMDs for the oral health-related quality of life, there is interest in implementing effective and safe interventions in this context [19]. Among these interventions, the injection of BoNTA in the masseter has been explored to manage myofascial pain [20,21,22], as well as sleep bruxism [23,24,25] and masseter hypertrophy [26]. In a published report, the unilateral injection of BoNTA in the quadriceps of young adult mice promoted atrophic changes in its target but also in a functionally and anatomically related non-treated muscle from the same injected side, the *gastrocnemius* [27]. Although mouse masticatory muscles work synchronously to control mandibular movements, this approach has not been used to understand how the BoNTA intervention in one single muscle affects its agonist and antagonist counterparts in the craniofacial region. Therefore, here, we aim to test the hypothesis that unilateral masseter hypofunction induces the increase in the gene expression of atrophy-related molecular markers (*Atrogin-1/MAFbx*, *MuRF-1,* and *Myogenin*), resulting in a 3D atrophic phenotype of the masseter and its agonist masticatory muscles (the temporalis and the medial pterygoid) from the ipsilateral side in adult mice.

## 2. Results

To analyze the changes in the molecular determinants of muscle atrophy (*Atrogin-1/MAFbx*, *MuRF-1*, *Myogenin*) in all the mice masticatory muscles, the relative mRNA levels were assessed using qRT-PCR at 2d, 7d, and 14d after unilateral BoNTA injection in the masseter. Figure 1 depicts the intra-individual changes in gene expression between the control side (saline-injected masseter) and the experimental side (BoNTA-injected masseter) in all the main masticatory muscles. In addition, Table 1 summarizes the descriptive statistics of the atrophy-related gene expression in masticatory muscles (mean ± SEM, *p*-value, confidence interval 95%, and Cohen’s dz effect size).

*Atrogin-1/MAFbx*, *MuRF-1,* and *Myogenin* mRNA gene expression was significantly increased at 2, 7, and 14d for both masseter (Figure 1A–C, Table 1) and temporalis (Figure 1D–F, Table 1) from the experimental side compared with the muscles from the control side. *Atrogin-1/MAFbx* and *MuRF-1* expression rose as early as 2d and tended to reduce at 14d, while *Myogenin* expression gradually rose from 2 to 14d, reaching a 70-fold increase in the masseter muscle. The high dispersion in the atrogene expression levels means that the CI95% includes the zero at some time points. However, in both masseter and temporalis, a high effect size is observed for all the genes and time points addressed (Cohen’s dz > 0.8, Table 1).

For the medial pterygoid, significant increases in the mRNA levels were only found at 7d for *Atrogin-1/MAFbx* and 2d–7d for *Myogenin*, with no changes at any time point for *MuRF-1* (Figure 1G–I, Table 1). On the other hand, the lateral pterygoid did not exhibit such changes in the gene expression of these atrophy-related molecular markers. It even displayed a reduced mRNA level for *MuRF-1* at 2d on the experimental side (Figure 1J–L). Although a significant *p*-value for *MuRF-1* was not detected in the lateral pterygoid with BoNTA, an increase was observed on the experimental side in three of the five animals from 2d and four of the six animals from 7d, in addition to medium–high Cohen’s effect sizes (Figure 1; Table 1). Interestingly, the timing of the increases in *Atrogin-1/MAFbx*, *MuRF-1,* and *Myogenin* in the lateral pterygoid resembles that observed in the masseter and temporalis muscles.

On the other hand, the lateral pterygoid did not exhibit an increase in the gene expression of these atrophy-related molecular markers on the experimental side. Furthermore, there was a significant reduction in the mRNA level for *MuRF-1* at 2d on the experimental side (Figure 1J–L; Table 1).

Both the volume and average thickness of the masticatory muscle volume are depicted in Figure 2 and Figure 3 (for the 3D depiction, see the Appendix A). The 3D atrophic phenotype of the masseter, temporalis, and medial pterygoid from the BoNTA-injected side is characterized by a smaller volume and reduced thickness relative to the same muscles from the saline-injected side. The latter is consistent with their 3D morphometric parameters (Table 2). In contrast, the lateral pterygoid from the BoNTA side exhibited a significant increase in volume and no changes in thickness at 14d (Figure 2 and Figure 3; Table 2). Interestingly, detailed phenotyping of the masticatory muscles using 3D thickness meshes revealed a regional increase in the anterior portion of the temporalis from the BoNTA side, which was consistent among all the samples (See Appendix A).

## 3. Discussion

In this study, we determined for the first time the expression pattern of atrophy-related molecular markers and the 3D phenotype of the main masticatory muscles in an adult mouse model of unilateral masseter hypofunction induced by a botulinum toxin type A (BoNTA) injection. Our experimental design involved a single BoNTA intervention in the masseter muscle on one side only, using the contralateral masseter as a control injection (injected with saline solution). At 2d, 7d, and 14d, the BoNTA-injected masseter, and its ipsilateral agonist muscles (i.e., the temporalis and the medial pterygoid) exhibited an atrophic phenotype when compared with the control side muscles. Therefore, our results support the hypothesis that the unilateral masseter hypofunction, induced by BoNTA, promotes an increase in the gene expression of atrophy-related molecular markers that precedes the 3D atrophic phenotype, not only in the BoNTA injected muscle but also in its agonists from the experimental side. Our phenotypic findings are thus in accordance with those observed in the postcranial muscles [27].

It is well known that denervation (either physically with surgery, or chemically induced by BoNTA) re-expresses *Myogenin* in adult skeletal muscle fibers. In murine models, when *Myogenin* increases in adult skeletal muscle fibers, it leads to the upregulation of the atrogenes *Atrogin-1/MAFbx* and *MuRF-1*, resulting in muscle atrophy [28,29,30]. The latter is opposite to the pro-myogenic role of *Myogenin* when expressed in satellite cells. In addition, *Myogenin* overexpression has also been observed in denervated human skeletal muscles [31]. Interestingly, *Myogenin* and atrogenes were overexpressed even in non-chemically denervated muscles, such as the temporalis and the medial pterygoid. We previously performed pilot experiments using Trypan Blue solution to determine that our injection did not diffuse to structures other than the target (i.e., the masseter). Additionally, a study in BALB/c mice demonstrated that BoNTA remains at the injection site when injected intramuscularly, unlike other routes of administration in which it diffuses into multiple tissues (i.e., oral, intraperitoneal, and intravenous) [32]. Therefore, it is unlikely to expect diffusion of BoNTA from the injected masseter to neighboring muscles. Moreover, no upregulation of atrogenes was observed in the lateral pterygoid from the experimental side. On the other hand, the biochemical crosstalk between the injected masseter muscle and the other masticatory muscles could have been mediated by secreted molecules [33]. Future work will be required to elucidate the influence of the different options mentioned.

Mice are suitable models for the study of mandibular dynamics [4]. We previously demonstrated that a single unilateral intramuscular injection of BoNTA in the masseter resulted in altered atrogene mRNA expression in this muscle in the experimental side compared with the control side (saline injected) at 7d [8]. In this study, we unveiled changes in the gene expression of atrophy-related molecular markers in the remaining non-injected masticatory muscles (i.e., temporalis, medial pterygoid, and lateral pterygoid) when unilateral masseter hypofunction was induced by BoNTA. Also, these changes preceded a 3D atrophic phenotype observed at 14d for the BoNTA-injected masseter and its non-injected agonists from the experimental side. Although the present study exhibits limitations, such as the use of only male animals and the lack of functional analysis of the masticatory muscles, our mouse model for BoNTA-induced unilateral masseter hypofunction appears to be highly reproducible and allows for the study of the cellular and molecular responses for soft and hard tissues under controlled conditions. Further research involving both sexes to improve musculoskeletal research [34] and a functional evaluation of the masticatory muscles is therefore recommended.

In mice, the masticatory muscles are in a close relationship, from an anatomical and functional perspective, with the functional musculoskeletal patterns similar to those described in humans [2,3]. The disturbance of the mechanical activity of the masseter muscle could modify the activity pattern of the other masticatory muscles, driving changes in the mandibular dynamics and, therefore, in the integration between masticatory muscle agonists and antagonists. Considering that the temporalis and the medial pterygoid are both agonists of the masseter muscle [3,35], our findings are consistent with the expected reduced masticatory activity induced by the unilateral BoNTA intervention. In a previous study, we determined that the experimental side, with masseter hypofunction, presented reduced masticatory activity, which was supported indirectly by the lack of enamel wear in the molar teeth [10]. This evidence also suggests that the reduced masticatory activity will negatively impact muscle function of the masseter agonists, inducing their atrophy, as shown by this study. Furthermore, a functional compensatory mechanism for reduced function of the masseter agonists may explain why the anterior portion of the temporalis exhibited an increase in thickness, and the lateral pterygoid exhibited an increase in volume on the experimental side. This functional hypothesis is supported by previous studies that evaluated the integration of the masticatory muscles in mice [1,5]. In particular, the lateral pterygoid, which is attached to the anterior portion of the mandible [3], is involved in the anterior displacement of the mandibular condyle on the balancing side during mastication [5]. Also, the relationship between less mechanical demand and reduced muscle structure has been described in adult humans [36]. Therefore, a reduction in the mastication process in the experimental side will increase this activity in the control side, demanding more activity of the lateral pterygoid, in contrast to the masseter agonists.

The biomechanical integration of the masticatory muscles during unilateral BoNTA-induced masseter hypofunction impacts structures beyond the skeletal muscles, including the subjacent mandibular bone [8,9,37]. Previous studies have demonstrated that masticatory muscle imbalance alters the shape of the craniofacial structures [9,33,38,39]. Although the shape changes in the mouse mandibular condyle after BoNTA-induced masseter hypofunction were only demonstrated at 14d, this is its earliest report compared with previous evidence [38,39,40]. Recently, a meta-analysis from a systematic review suggested that bone loss has been mainly evaluated in the mandibular condyle in animal studies. In contrast, human studies are focused on the mandibular ramus (where the masseter muscle is inserted) [41]. However, due to a low number of human studies and small size samples, combined with short-term follow-ups and the lack of dose standardization, there is a need for more clinical research to determine if these bone changes may negatively impact structures such as the temporomandibular joint, which is vital for physiological processes [40,41]. A clinical study demonstrated that using the BoNTA intervention to manage myofascial pain in adult women resulted in thickness reduction of the injected muscles (both masseter and temporalis), and reduced bone volume in the mandibular condyle in a dose-dependent manner; nonetheless, non-injected muscles were not assessed [42]. Thus, more clinical studies are required to determine the intended and side effects of this procedure on the human population.

On the other hand, there is a limited comprehension of the cellular and molecular mechanisms behind bone loss during botulinum toxin-induced skeletal muscle atrophy [37,40]. Therefore, our experimental mouse model is relevant for understanding further the crosstalk between muscle and bone beyond mechanical relationships and how the impact of interventions performed in clinical dentistry, such as BoNTA intramuscular injection, particularly in the adult human population, can be potentially used to manage several craniofacial conditions. Finally, we also propose that the musculoskeletal relationship should be considered to also involve molecular factors, which remains to be explored in the craniofacial system [33,43].

## 4. Materials and Methods

### 4.1. Unilateral Masseter Hypofunction: Adult Mouse Model

Twenty-three 8-week-old male BALB/c mice were obtained from the animal research facility (Faculty of Dentistry, Universidad de Chile). The intervention was performed as previously reported [8,9]: each animal received a single injection of botulinum toxin type A (BoNTA; 0.2U/10 µL; Onabotulinumtoxin A; BOTOX^®^, Allergan, North Chicago, IL, USA) at baseline in the right masseter (experimental side), whereas the left masseter received the same volume of saline solution (0.9% *w*/*v* NaCl; control side). All animals were maintained under standardized temperature and humidity conditions, with water and food (LabDiet^®^ JL Rat and Mouse/Auto 6F 5K67; LabDiet, St. Louis, MO, USA) ad libitum. Intraperitoneal overdose of anesthesia was used for euthanizing the animals at 2d (*n* = 6), 7d (*n* = 6), and 14d (*n* = 6). Masticatory muscles (the masseter, the temporalis, the medial pterygoid, and the lateral pterygoid) were harvested and stored at −80 °C until further processing. In addition, five complete heads (with the masticatory muscles in situ) were isolated at 14d, dissected to remove the skin, and stored in 10% neutral buffered formalin solution (Sigma Aldrich^®^, St. Louis, MO, USA) at room temperature until processing. All procedures were approved by the CICUA (Institutional Animal Care and Use Committee) of Universidad de Chile (certificate No 17011-OD-UCH) and performed following the institutional guidelines and the ARRIVE guidelines (Animal Research: Reporting In Vivo Experiments). Sample size calculation was performed with G*Power 3.1.9.7 [44], considering an α error of 0.05 and a power (1-β) of 0.8. Effect size obtained in a pilot study for atrogene expression in masseter muscle (1.648) or masseter mass reduction (dz = 1.765) after BoNTA injection (7–14 d, respectively) provided a sample size of 6 for mRNA assays and 5 for morphological changes detection.

### 4.2. Relative Levels of mRNA of Atrophy-Related Molecular Markers in Mouse Masticatory Muscles

Masticatory muscles from treated mice at 2d, 7d, and 14d were processed individually with mechanical homogenization using Trizol™ (Invitrogen, Thermo Fisher Scientific, Waltham, MA, USA). Total RNA was extracted following manufacturer’s instructions. cDNA was generated from 1 µg of RNA with the High-Capacity cDNA Reverse Transcription Kit (Applied BiosystemsTM, Thermo Fisher Scientific, MA, USA), followed using the DNA-free™ DNA Removal Kit (InvitrogenTM, Thermo Fisher Scientific, Waltham, MA, USA).

To quantify the relative mRNA expression of atrophy-related molecular markers (*Atrogin-1/MAFbx*, *MuRF-1,* and *Myogenin*) in mouse masticatory muscles, quantitative real-time PCR (qRT-PCR) was performed using previously reported primers [45], as follows: *Atrogin-1/MAFbx* forward: 5′-GTTTTCAGCAGGCCAAGAAG-3′, reverse: 5′-TTGCCAGAGAACACGCTATG-3′; *MuRF-1* forward: 5′-TGCCTACTTGCTCCTTGT-3′, reverse: 5′-CTGGTGGCTATTCTCCTT-3′; *Myogenin* forward: 5′-CTCCCTTACGTCCATCGT-3′, reverse: 5′-CAGGACAGCCCCACTTAA-3′ [46]. *GAPDH* was used as housekeeping gene: forward: 5′-CAACTTTGGCATTGTGGAAG-3′, reverse: 5′-CTGCTTCACCACCTTCTTG-3′ [47]. qPCR was performed as previously described [8,9]. 2^−ΔΔCT^ ± standard error of the mean (SEM) units were used for reporting the gene expression [48].

### 4.3. The Three-Dimensional Phenotyping of Mouse Masticatory Muscles with X-ray Microtomography (µCT)

To allow for the detection of masticatory muscles using µCT, soft tissue staining for contrast-enhanced imaging was performed. Complete heads from 5 animals isolated at 14d were fixed in 10% neutral buffered formalin solution for at least 48 h and then washed under running water overnight. Dehydration of samples was achieved using serial immersion in alcohol: 30% for 2 h, 50% for 2 h, and finally stored in 70% ethanol. Phosphotungstic acid (PTA) powder (Sigma Aldrich^®^, St. Louis, MO, USA) was reconstituted in 70% ethanol (0.025 g/mL) and each sample was immersed in 250 mL of 2.5% PTA solution for two weeks under continuous shaking. Samples were stored in 70% ethanol until scanning.

To determine the 3D phenotype of the mouse masticatory muscles in situ at 14d, PTA-stained complete head samples were scanned with a DIONDO d3 µCT device (DIONDO GmbH, Hattingen, Germany), while immersed in 70% ethanol, following these parameters: voltage 130 kV, current 40 mA, 0.5 mm filter (Brass), integration time 1500 ms and an isotropic voxel size of 5.5 µm. After reconstruction, large volumetric µCT datasets of TIFF (Tagged Image File Format) files were down-sampled by 4-fold in X, Y and Z planes, maintaining the isotropy of the voxels, with ImageJ 1.52e [49]. Scans were pre-visualized in Avizo 9.2 (Thermo Scientific™, Waltham, MA, USA) (See Appendix A) and a single operator (JB-M) pre-segmented the masticatory muscles in coronal views every 20 slices (as per protocol, remaining slices must be left unsegmented). Then, original and labeled datasets were processed in the online platform of the Biomedical Image Segmentation App (Biomedisa) [50,51] for muscle segmentation. Output files from Biomedisa were uploaded onto DRAGONFLY 2021.3 (Object Research Systems, Montreal, Canada), and each muscular group was identified and color-coded based on the obtained histogram after segmentation (See Appendix A). Morphometric parameters (the volume and the average thickness from all slices) from each muscle were obtained from 3D geometries. A 3D rendering video registering 3D muscles on a representative sample of a skull and a mandible, manually obtained by a single operator (AE-R) from a down-sampled µCT dataset, was produced using DRAGONFLY 2021.3 (Movie Maker tool) at 30 FPS and processed using iMovie for Mac 10.3.1. (See Appendix A).

### 4.4. Statistical Analysis

Results are shown as mean ± standard error of the mean (SEM). Paired analysis (the Wilcoxon signed-rank test) was used for gene expression and morphometric masticatory muscle parameters between sides. The non-parametric test was chosen due to the small sample size (5–6). GraphPad Prism for Mac 9.3.1 (GraphPad Software, La Jolla, CA, USA) was used for all the statistical analyses. A *p* < 0.05 was set up to determine statistical significance. For gene expression, 95% confidence intervals (CI) were calculated for intra-individual differences (BoNTA side and saline side). Cohen’s dz effect size was calculated with G*Power 3.1.9.7 [44], as post-hoc, using the mean and the standard deviation as input, based on a Wilcoxon signed-rank test (matched pairs).

## 5. Conclusions

We conclude that the unilateral BoNTA-induced masseter hypofunction in adult male mice increases the mRNA expression of atrophy-related molecular markers in the BoNTA-injected muscle and in the masseter-agonist non-injected masticatory muscles from the experimental side, at 2d, 7d, and 14d after intervention. This pattern in gene expression results in a 3D atrophic phenotype of the same muscles, characterized by the smaller volume and reduced average thickness compared to those from the control side, evaluated at 14d in situ.

## Figures and Tables

**Figure 1 ijms-24-14740-f001:**
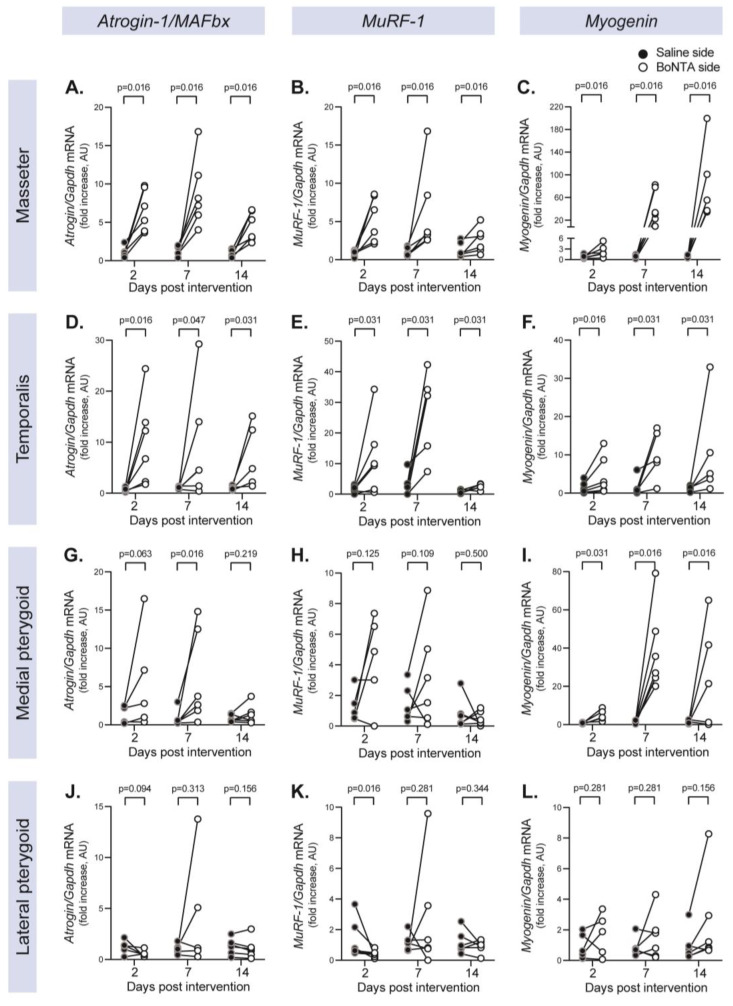
Unilateral masseter hypofunction increases the gene expression of atrophy-related molecular markers in its agonist masticatory muscles from the ipsilateral side. Mice masseter muscles were injected with BoNTA (right side) and saline solution (left side). At different days post-intervention, the following masticatory muscles were isolated and processed for detection of atrogenes (*Atrogin-1/MAFbx*, *MuRF-1*, *Myogenin*) using qRT-PCR: masseter (**A**–**C**), temporalis (**D**–**F**), medial pterygoid (**G**–**I**), lateral pterygoid (**J**–**L**). Values are normalized with *Gapdh* and expressed as fold increase in the saline side for each condition (muscle and time point) (2^−ΔΔCt^ method). Results are shown as paired data connecting the saline side (black circles) and the BoNTA side (white circles) of each animal with a line. *p*-values were obtained with the Wilcoxon paired one-tailed test, BoNTA vs. saline side for each muscle and time point; *n* = 5–6.

**Figure 2 ijms-24-14740-f002:**
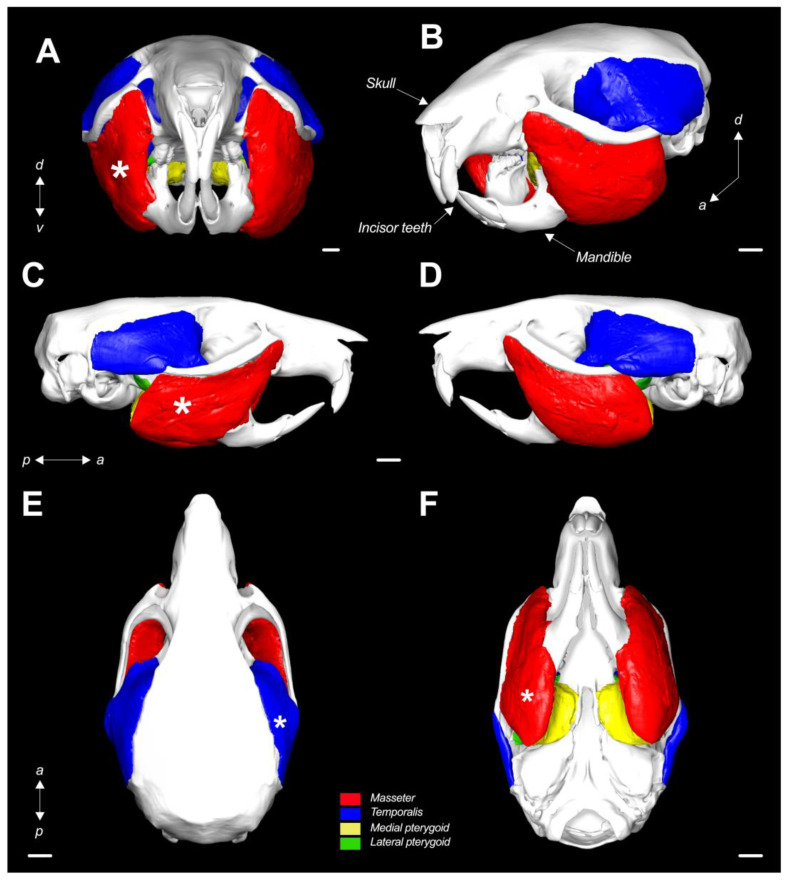
Three-dimensional rendering of the masticatory muscle volume in the adult mouse head at 14d (in situ). (**A**) Frontal view of representative sample exhibiting atrophic phenotype of both masseter and temporalis from the BoNTA-injected side, characterized by smaller volume compared with muscles from saline-injected side; (**B**) rotated (after pitch, roll, and yaw) view of muscle volume registered on segmented bones (skull and mandible) from the same sample; (**C**) BoNTA-injected side and (**D**) saline-injected side sagittal views; (**E**) dorsal and (**F**) ventral views. White asterisks: BoNTA-injected side. Abbreviations: *d*, dorsal; *v*, ventral; *a*, anterior; *p*, posterior; scale bar (white line): 1 mm.

**Figure 3 ijms-24-14740-f003:**
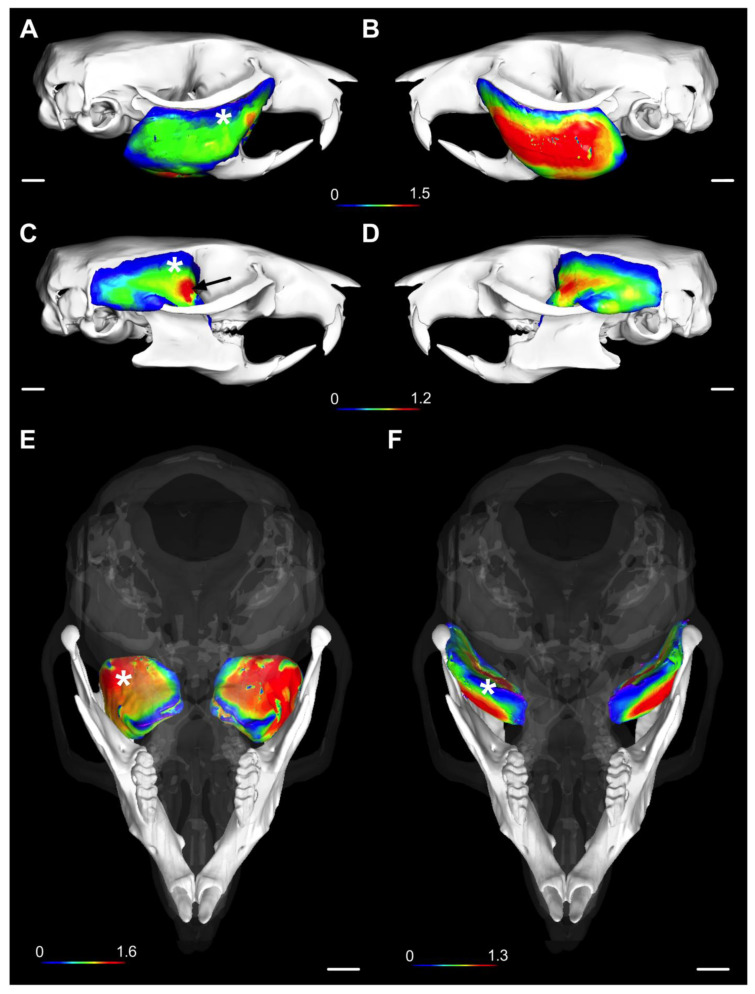
Three-dimensional rendering of the masticatory muscle thickness in the adult mouse head at 14d (in situ). (**A**,**B**) Sagittal views of masseter from (**A**) BoNTA-injected side and (**B**) saline-injected side. (**C**,**D**) Sagittal views of temporalis from (**C**) BoNTA-injected side and (**D**) saline-injected side. In (**C**), a regional increase in thickness in the anterior muscle fascicles of the temporalis from the BoNTA side can be identified (black arrow), compared with its counterpart from the saline side. (**D**) Cranial (anterior) views of (**E**) medial and (**F**) lateral pterygoid (with transparent skull), showing only different phenotypes for thickness distribution in the medial pterygoid from the BoNTA side, compared with the saline side. White asterisks: BoNTA-injected side; color-coded bar in mm; scale bar (white line): 1 mm.

**Table 1 ijms-24-14740-t001:** Descriptive statistics of the atrophy-related gene expression in masticatory muscles of the control or BoNTA-injected side.

						BoNTA Side–Control Side	
Muscle	Days	Gene	Control Side Mean ± SEM	BoNTA Side Mean ± SEM	*p* Value *	CI 95% **	Cohen’s dz ***
Masseter	2	Atrogin-1	1.15 ± 0.28	6.55 ± 1.11	0.016	[2.363, 8.429]	1.867
		Murf1	0.82 ± 0.17	5.26 ± 1.21	0.016	[1.278, 7.612]	1.473
		Myogenin	0.79 ± 0.20	2.36 ± 0.68	0.016	[−0.181, 3.331]	0.941
	7	Atrogin-1	1.15 ± 0.25	8.88 ± 1.86	0.016	[2.611, 12.840]	1.585
		Murf1	1.02 ± 0.23	6.24 ± 2.30	0.016	[−0.678, 11.120]	0.929
		Myogenin	0.63 ± 0.20	42.72 ± 12.35	0.016	[10.220, 73.950]	1.386
	14	Atrogin-1	0.84 ± 0.21	4.49 ± 0.78	0.016	[1.535, 5.754]	1.813
		Murf1	0.80 ± 0.31	2.54 ± 0.69	0.016	[0.696, 2.775]	1.751
		Myogenin	0.87 ± 0.12	77.13 ± 26.59	0.016	[7.825, 144.700]	1.169
Temporalis	2	Atrogin-1	0.68 ± 0.16	10.22 ± 3.49	0.016	[0.845, 18.250]	1.151
		Murf1	1.72 ± 0.45	12.01 ± 5.04	0.031	[−1.640, 22.210]	0.906
		Myogenin	1.37 ± 0.62	4.67 ± 2.06	0.016	[−0.410, 7.024]	0.934
	7	Atrogin-1	1.02 ± 0.09	9.05 ± 4.49	0.047	[−3.473, 19.540]	0.732
		Murf1	3.13 ± 1.78	26.37 ± 6.41	0.031	[4.173–42.320]	1.514
		Myogenin	1.61 ± 1.13	10.11 ± 2.87	0.031	[−0.165, 17.170]	1.218
	14	Atrogin-1	1.07 ± 0.18	7.21 ± 2.77	0.031	[−1.187, 13.470]	1.041
		Murf1	0.78 ± 0.27	2.41 ± 0.44	0.031	[0.9810, 2.287]	3.107
		Myogenin	1.24 ± 0.20	10.70 ± 0.68	0.031	[−5.984, 24.900]	0.760
Medial pterygoid	2	Atrogin-1	1.52 ± 0.50	5.56 ± 2.98	0.063	[−3.262, 11.340]	0.687
		Murf1	1.28 ± 0.47	4.35 ± 1.32	0.125	[−0.811, 6.957]	0.982
		Myogenin	0.66 ± 0.18	4.89 ± 1.28	0.031	[0.5235, 7.918]	1.417
	7	Atrogin-1	0.83 ± 0.44	5.98 ± 2.49	0.016	[−0.634, 10.930]	0.935
		Murf1	1.38 ± 0.49	3.21 ± 1.35	0.109	[−1.302, 4.949]	0.612
		Myogenin	1.28 ± 0.34	39.23 ± 9.00	0.016	[5.120, 60.790]	1.744
	14	Atrogin-1	0.81 ± 0.20	1.35 ± 0.52	0.219	[−0.619, 1.699]	0.489
		Murf1	0.79 ± 0.42	0.49 ± 0.19	0.500	[−1.725, 1.122]	0.302
		Myogenin	1.23 ± 0.39	25.89 ± 12.39	0.016	[−10.50, 59.82]	0.870
Lateral pterygoid	2	Atrogin-1	1.23 ± 0.31	0.59 ± 0.15	0.094	[−1.674, 0.394]	−0.769
		Murf1	1.38 ± 0.52	0.41 ± 0.10	0.016	[−2.327, 0.390]	−0.748
		Myogenin	0.83 ± 0.33	1.42 ± 0.57	0.281	[−0.838, 2.001]	0.43
	7	Atrogin-1	1.05 ± 0.22	4.23 ± 2.53	0.313	[−3.831, 10.190]	0.563
		Murf1	1.11 ± 0.24	2.68 ± 1.47	0.281	[−2.589, 5.272]	0.396
		Myogenin	0.86 ± 0.25	1.56 ± 0.63	0.281	[−0.967, 2.377]	0.443
	14	Atrogin-1	1.29 ± 0.33	1.12 ± 0.41	0.156	[−0.558, 0.2178]	−0.460
		Murf1	1.18 ± 0.31	0.90 ± 0.17	0.344	[−1.060, 0.503]	−0.367
		Myogenin	0.96 ± 0.42	2.45 ± 1.21	0.156	[−1.905, 4.902]	0.462

Control side: left side injected with saline solution only in the masseter muscle; BoNTA side (experimental): right side injected with BoNTA only at the masseter muscle. Expression of atrogenes (Atrogin-1, Murf1, Myogenin) was addressed in all the main masticatory muscles at different time points after intervention (2, 7, 14d) using RT-qPCR. All the values were normalized with the housekeeping GAPDH and expressed as fold increase relative to the saline-injected side for each condition (muscle and time point) using the 2^−ΔΔCt^ method. *n* = 5–6 mice per time point. * *p*-value calculated using the Wilcoxon paired one-tailed test, BoNTA- vs. control side for each muscle and time point. *p* < 0.05 was considered significant and is highlighted in gray color. ** 95% lower and upper confidence intervals (CI) were calculated for differences (BoNTA side and control side). *** Differences (BoNTA side–control side) of mean and standard deviation were input for Cohen’s dz effect calculation. G-Power 3.1.9.7, post-hoc, Wilcoxon’s signed-rank test (matched pairs).

**Table 2 ijms-24-14740-t002:** The three-dimensional morphometric parameters of masticatory muscles.

	Volume (mm^3^)			Thickness (mm)		
Muscles	Control Side Mean ± SEM	BoNTA Side Mean ± SEM	Mean Difference	Control Side Mean ± SEM	BoNTA Side Mean ± SEM	Mean Difference
Masseter	43.6 ± 1.3	27.4 ± 0.4	−16.3 *	0.87 ± 0.01	0.64 ± 0.01	−0.23 *
Temporalis	21.3 ± 0.6	18.6 ± 0.7	−2.6 *	0.57 ± 0.00	0.56 ± 0.00	−0.01 *
Medial pterygoid	13.7 ± 0.6	11.5 ± 0.5	−2.2 *	0.97 ± 0.02	0.88 ± 0.02	−0.09 *
Lateral pterygoid	7.4 ± 0.2	7.9 ± 0.2	+0.5 *	0.72 ± 0.01	0.73 ± 0.01	+0.01 ^ns^

Control side: left side injected with saline solution only at the masseter muscle; BoNTA side (experimental): right side injected with BoNTA only at the masseter muscle. *n* = 5; *: *p* < 0.05; ns, non-significant difference; Wilcoxon’s paired one-tailed test, experimental vs. control side for each muscle at 14d.

## Data Availability

Not applicable.

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
