# Peer review of "Unilateral Hypofunction of the Masseter Leads to Molecular and 3D Morphometric Signs of Atrophy in Ipsilateral Agonist Masticatory Muscles in Adult Mice"

_ijms, 2023, doi:10.3390/ijms241914740_

Round 1

Reviewer 1 Report

The manuscript of Balanta-Melo et al. investigates the influence of botulinum toxin type A injection in the masseter on the ipsilateral agonist masticatory muscles. The authors measured muscular atrophy determinants (Atrogin-1/MAFbx, MuRF-1, Myogenin) mRNA levels in M. masseter, M. temporalis, and medial pterygoid. Muscle thickness has been measured and modeling in 3D has been performed. The authors confirmed the atrophic phenotype of not-injected, ipsilateral muscles and, given that botulinum toxin diffusion and denervation mechanisms seem to be not probable, hypothesize that other mechanism could be involved in the atrophy.

The introduction provides a sufficient background on the mice model of the mandibular movement, BoNTA applications for the management of clinical dentistry and previous studies on this topic.

Important limitations regarding the mechanical activity of the masseter muscle that could modify the activity pattern of the other masticatory muscles are highlighted. Tables and figures are of good quality, but 3D modeling and video would need some additional explanation regarding the localization of anatomical structures. 

Comments:

  1. The vocabulary should be more sensitive: "sacrificed" instead of e "euthanized".Please avoid the expression "heads were harvested"
  2. For the video and Figure 2 and figure 3 it would be good to put some arrows and name the anatomical structures.
  3. How were the genes for mRNA expression chosen? Was it not possible to do RNA-Seq?
  4. If denervation and diffusion seem to be improbable, what other mechanisms may be involved in the atrophy?
  5. Have  any pathomorphological investigations been performed (staining, microscopy )? 
  6. What would be a potential practical influence of this study on humans? Please provide examples if possible.

Reviewer 2 Report

Review of an article titled ‘’ Unilateral hypofunction of the masseter leads to molecular and 3D-morphometric signs of atrophy in ipsilateral agonist masticatory muscles in adult mice’’.

The study is interesting, congratulations on the idea. In my opinion, the introduction should be developed and the statistical analysis should be elaborated in more detail. Detailed comments below.

Abstract

1.      L40 - In my opinion, it must be explicitly written that the test should be repeated on humans to verify the results.

Introduction  

2.      What I miss in the introduction is why it is so important to test the masticatory muscles. Why their proper physiology is important for human well-being. Also mark the epidemiological data of the most common disease entity associated with the masticatory muscles - TMDs. According to the WHO, it is the most common third dental disease.

Remember that the masticatory muscles are susceptible to changes related to factors such as psychological stress, numerous studies link temporal myalgia to tension type headaches, viral infections (e.g. COVID-19), some studies link muscle changes to wearing masks, other studies link the effect of visual impairment on the masticatory muscles.

To summarize my point, add a paragraph about why masticatory muscles and their diagnosis and treatment are so important for people.

3.      L69-82- In the last paragraph of the introduction, please clearly write the aim of the study.

Results

4.      Beforehand, read my comments in the Materials and Methods section. Add exact p-score values with effect size and confidence interval. 

Discussion

5.      L187 – ‘’ In mice, the masticatory muscles are in close relationship from an anatomical and functional perspective [3]’’ - In my opinion, the sentence is to be removed. In any, living mammal the masticatory muscles are in close relationship from an anatomical and functional perspective.

6.      Add pragraf about the limitations of the study.

I declare that I, again, will review in detail the discussions after improving the methods and results section. 

Materials and Methods

7.      L222 – ‘’ Thirty-one’’  - Where exactly did this number of animals come from? Present and add sample size calculations in the text.

8.      L222 – Justify in the text why only male mice were chosen.

9.      L230 – ‘’ 2d (n=10), 7d (n=10) and 14d (n=6).’’ - Please explain because I don't understand. In line 222 it says 31 mice here it says 26 mice. Where do these differences come from? 

10.   L232 and 258 – ‘’ five complete heads ‘’ and ‘’Complete heads from 5 animals harvested’’ - Again, why exactly 5 ? Present and add sample size calculations in the text.

Statistical analysis

11.   I understand that a non-parametric test was chosen because of the small number of individuals.  This is understandable and ok however for a person familiar with statistics. Your work will be read by people with different knowledge, so add a rationale for choosing a non-parametric test.

12.   Providing the ''p'' value alone is already insufficient. Obligatorily add the effect size. In addition, provide the formula from which you will count it. I am aware that this will be a challenge for Wilcoxon signed-rank test.

13.   Also, add a 95% confidence interval, then your statistic will be unassailable.

 Conclusions

14.  L229 - In my opinion, it must be explicitly written that the test should be repeated on humans to verify the results.

Minor comments

15.   L44 – ‘’ 1. Introduction’’ - It should not be left on the page. Move to the next page.

16.   Figure – To all figures - add information that ''*'' stands for the place of injecting.

17.   L254 –‘’ described (8).’’  - Incorrect form of quotation.

Thank you for the opportunity to review. With best regards.

Round 2

Reviewer 1 Report

Dear authors, 

Thank you for providing an exhaustive response. 

I believe that the manuscript is correct and well-written. However, I think it may be a better fit for a more specialized journal. This decision is up to the editor.

Author Response

Thank you for your comments. We consider that the manuscript involves several fields, such as dentistry, morphology, physiology, and cellular/molecular biology. So, IJMS will be a great platform to share our work. Even more in the special issue "Translational Myology: Cellular, Genetic, Molecular Aspects"

Reviewer 2 Report

Dear Authors, thank you for your responses and for correcting the manuscript. After reviewing your responses, most accept with the exception of my last comment #2.

Please respond appropriately and add the missing information. To make it easier for you, I  send you sample studies:

·        Epidemiological data  DOI: 10.1007/s00784-020-03710-w

·        Potential impact of pandemic on the severity of TMDs: doi: 10.7759/cureus.28167 ;  DOI     10.3390/ijerph192315559  doi: 10.3390/brainsci13030481

I think that the addition of this type of information will increase interest in your research. I think that adding this type of information will increase the importance of the TMDs problem and the significance of your results.

Additional comment: L75 – ‘’ TMD’’  - Should be TMDs - the acronym refers to a set of symptoms.

books/NBK557995/

Author Response

- Thanks for the input. In the new version of the manuscript we have incorporated in the third paragraph of the introduction the high prevalence of TMDs and the current interest in the topic due to its post-COVID increase, as follows (PMID is detailed for new references added):

“The mechanism of action of BoNTA, and its impact on skeletal muscle function and shape has gained attention in clinical dentistry to manage pathological and aesthetic craniofacial conditions [11]. Temporomandibular disorders (TMDs) are the most prevalent musculoskeletal diseases that affect the masticatory system, compromising vital functions such as mastication and speaking, at physical and psychosocial levels (12-13, PMID32200600), reaching as high as 31% in adult population (PMID: 33409693). Even more, due to the COVID-19 pandemic, there was an increase in the prevalence of TMDs among patients affected by this respiratory disease (PMID: 36158329- 36979291). Importantly, public health strategies to control COVID-19 spread, such as the use of surgical masks by the overall population, may impact the electromyographic activity of masticatory muscles such as the masseter and the temporalis, with differences between healthy patients and those affected by TMDs (PMID: 36497634). Particularly, electromyography of masticatory muscles has been proposed as a potential diagnostic tool to identify pain-related TMDs such as myofascial pain [13]. Thus, considering the benefits of managing painful TMDs on the oral-health-related quality of life, there is an interest on implementing effective and safe interventions on this context ( PMID: 37078711). Among these interventions, the injection of BoNTA in the masseter has been explored to manage myofascial pain [20-22], as well as sleep bruxism [23-25] and masseter hypertrophy [26].”

- TMD was changed to TMDs in the new version of the manuscript, as suggested.